



**Marine snow surface production and bathypelagic export at the**
**Equatorial Atlantic from an imaging float**
Joelle Habib[1], Lars Stemmann[1,6], Alexandre Accardo[1], Alberto Baudena[1], Franz Philip Tuchen[2,3], Peter Brandt[4,5],
Rainer Kiko[1,4,5]
[1]Sorbonne Université, CNRS, Laboratoire d'Océanographie de Villefranche, LOV, 06230 Villefranche-sur-Mer,
France
[2]Cooperative Institute for Marine and Atmospheric Studies, Rosenstiel School of Marine, Atmospheric, and Earth
Science, University of Miami, Miami, FL, USA
[3]NOAA/Atlantic Oceanographic and Meteorological Laboratory, Miami, FL, USA
[4]GEOMAR Helmholtz Centre for Ocean Research Kiel, Kiel, Germany
[5]Faculty of Mathematics and Natural Sciences, Kiel University, Kiel, Germany
[6]Institut Universitaire de France (IUF), Paris, France
*Correspondence to*: Joelle Habib (joellehabib22@hotmail.com)
**Abstract.** The marine biological carbon pump (BCP) plays a central role in the global carbon cycle, transporting
carbon from the surface to the deep ocean and sequestering it for long periods. Sinking of surface-produced
particles, known as the Biological Gravity Pump (BGP) constitutes the main component of the BCP. To study the
BGP in the equatorial Atlantic upwelling region, a biogeochemical (BGC) Argo float equipped with an Underwater
Vision Profiler 6 (UVP6) camera was deployed from July 2021 to March 2022. The float was recovered after its
eastward drift from 23°W to 7°W along the equator, during which it conducted profiles to 2000 m depth every
three days. For the first time in this oceanic region, in situ images and physical and biogeochemical data from a
BGC-Argo float were acquired and analyzed in combination with satellite data. During the float trajectory, two
blooms were recorded followed by two main export events of sinking aggregates that lasted for over a month,
consistently reaching 2000 m depth. A Lagrangian approach was applied to investigate the production,
transformation, and deep export of marine particles. Based on the characterization of the morphology of detritus
within and outside of the plumes, five particle morphotypes with different sinking properties were detected. Small
and dense aggregates were present throughout the water column while porous morphotypes, despite being larger,
were predominantly concentrated in the surface layer. Export was driven by small and compact particles with higher
particle abundance and flux during upwelling and export events. Our investigation reveals the stability of the
equatorial Atlantic BCP system during this period, yielding an export efficiency of 6-7% during and outside of
export events. This study highlights the importance of using new technologies on autonomous platforms to
characterize the temporal variability in the magnitude and functioning of the BCP.
**1 Introduction**
The term "biological carbon pump" (BCP) encompasses physical and biological processes responsible for the
generation, export, and remineralization of organic matter from the upper ocean to depth (Boyd et al., 2019;
DeVries et al., 2012; Steinberg & Landry, 2017). The biological pump connects various aspects of the carbon cycle:
the upper-ocean photosynthetic carbon uptake, the alimentation of the midwater biota (Irigoien et al., 2014), and
the carbon storage within the deep sea (Buesseler et al., 2007). Within the euphotic zone, organic particles are
continuously generated and recycled, with only a small fraction descending into deeper layers (De La Rocha, 2004),
while remineralization occurs within a few hundred meters of the surface and is facilitated by processes such as



zooplankton feeding or microbial degradation (Giering et al., 2014; Steinberg & Landry, 2017; Stemmann, Jackson,
& Ianson, 2004). Given the retroactive potential of the BCP to significantly impact anthropogenic climate warming
(Bernardello et al., 2014; Bopp et al., 2013), understanding the multitude of mechanisms governing the BCP is of
paramount importance.
Among the different processes of the BCP, sinking marine snow is the key component of particulate carbon
transport to the deep ocean, a process known as the biological gravitational pump (BGP). Marine snow consists of
detritus, formed from a mixture of source particles produced by the surface ecosystem and aggregated together by
physical (coagulation) or biological (trophic activity) mechanisms (Alldredge & Silver, 1988). Their composition
is determined by multiple characteristics, mainly the phytoplankton and zooplankton community composition
(Bach et al., 2019; Tréguer et al., 2018). In the mesopelagic layers, several biological and physical factors influence
their dynamics and control their size distribution and morphology, which affect their sinking (Cael et al., 2021;
Stemmann, Jackson, & Ianson, 2004). Shear and differential settling modulates aggregation (Jackson, 1990;
Stemmann et al., 2004) while fragmentation rates have been proposed to depend on shear and swimming organisms
(Briggs et al., 2011; Dilling & Alldredge, 2000; Jackson, 1990). Additionally, particle volume and surface area
condition interactions with microorganisms (e.g., colonization and degradation of particles; Bianchi et al., 2018),
modifying marine snow morphology by making them more porous and fragile with time (Biddanda & Pomeroy,
1988; Ploug & Grossart, 2000).
An efficient tool to track the particle morphology, study their abundance, and estimate the vertical carbon flux is
the Underwater Vision Profiler (UVP; Picheral et al., 2010, 2022). This imaging tool measures particle abundance
and distribution (Guidi et al., 2009; Kiko et al., 2022; Stemmann et al., 2002), to estimate the biological
gravitational pump (Forest et al., 2013; Guidi et al., 2015; Kiko et al., 2017; Ramondenc et al., 2016), and more
recently, to explore particle morphology (Trudnowska et al., 2021; Accardo et al., submitted). One decade of
observation with the UVP5 during ship surveys allowed a global monitoring (Forest et al., 2013; Guidi et al., 2008,
2015; Kiko et al., 2017; Stemmann et al., 2002) and enabled the reconstruction of global export fluxes from the
spatially variable euphotic zone and mixed layer depths (Clements et al., 2022, 2023; Guidi et al., 2015).
Despite significant improvement in observation capacities from ships, high frequency observations during long-
term deployment to study relevant scales of marine snow dynamics over a large depth range was not possible.
Autonomous platforms equipped with imaging sensors have emerged and are currently being utilized to remotely
record plankton and particle distributions in addition to the core parameters such as salinity, temperature, and
optically derived other variables (Claustre et al., 2020; Picheral et al., 2022). Recently, surface blooms followed
by plumes of sinking material were monitored using optical sensors (fluorescence and backscatter) mounted on
BGC-Argo float drifting in a quasi-Lagrangian mode (Briggs et al., 2011, 2020) and global POC standing stocks
have been calculated (Fox et al., 2024). Such studies with optical sensors (fluorescence, backscatter) are key to
understanding particle dynamics in the core of the oceans but they are not adapted to study marine snow.
We selected the equatorial Atlantic Ocean to conduct our study, as it is characterized by enhanced primary
productivity concentrated within the equatorial and coastal regions (Grodsky et al., 2008). This productivity is due
to the presence of upwelling zones in the central and eastern parts of the equatorial basin (Schott et al., 1998) which
bring nutrients to the euphotic zone (Radenac et al., 2020). This enhanced productivity results in a stronger passive



and active export of particulate matter reaching up to 4000 m (Kiko et al., 2017). The strength of the equatorial upwelling system is modulated by the strength of seasonally varying winds associated with the meridional migration of the intertropical convergence zone (Brandt et al., 2023). At intraseasonal (20-50 days) scales, Tropical Instability Waves (TIWs) are another factor influencing the equatorial local productivity. TIWs are westward-propagating, cusp-shaped oscillations prevalent in the central and western equatorial Atlantic generated by baroclinic and barotropic instabilities (Athie & Marin, 2008). They induce strong intraseasonal variations in sea surface temperature, sea surface salinity, and ocean currents (Tuchen et al., 2022), and are associated with sharp fronts (Warner et al., 2018). TIWs can also influence nitrate (Radenac et al., 2020) and chlorophyll distribution (Menkes et al., 2002; Sherman et al., 2022).

We here focus on the equatorial Atlantic BGP, using data from a UVP6 camera mounted on a BGC-Argo float deployed at 23°W, 0° in July 2021 to study the impact of seasonal upwelling and intraseasonal TIWs on productivity and particle export. In particular, we use a plume-based approach to follow the initiation and vertical extent of export events, to characterize particle production of various morphotypes during two bloom events, and to describe the patterns of their attenuation as they are exported to the meso- and bathypelagic layers.

## 2 Material and Methods

### 2.1 Satellite data

#### 2.1.1 Sea surface chlorophyll-a

Estimates of chlorophyll-a (chl-a) concentration and anomalies for the tropical Atlantic were obtained from the combination of two different products: the Global Ocean Color product (OCEANCOLOUR_GLO_BGC_L4_MY_009_104) produced by ACRI-ST and the NOAA-VIIRS provided by NOAA CoastWatch. Both of these data sets provide gap-free time series, with a temporal extent from 1997 till 2023 for the first product, while the second one only started in 2018. The temporal resolution for both products is one day with a spatial resolution of 4 km for the first product and 9 km for the second one.

#### 2.1.2 Sea surface temperature

Sea surface temperature (SST) and SST anomaly data were downloaded from the NOAA OI-SST data set (Huang et al., 2021; https://psl.noaa.gov/data/gridded/data.noaa.oisst.v2.highres.htm). SST anomalies are computed relative to a 30-year climatological mean. The gridded data are available daily from 1981-present at a horizontal resolution of 0.25°. To isolate TIW induced SST variability from the time series, a temporal (20-50 days) and a zonal (4-20° wavelength) bandpass filter were applied in accordance with previous studies (Olivier et al., 2020; Tuchen et al., 2022).

#### 2.1.3 Lagrangian diagnostics

Several Lagrangian diagnostics were computed for each sampling station using velocity data and environmental satellite products. To this aim, we defined for each station a circular region that we consider representative of the water parcel sampled by the BGC-Argo float. A radius of 0.1° was used (consistently with previous studies, Baudena et al., 2021; Fabri-Ruiz et al., 2023; Ser-Giacomi et al., 2021), and the circular region was filled with virtual particles. A given diagnostic is calculated for each virtual particle in the circular region. These values are then averaged together, providing one value of a given diagnostic per station.



The velocity field used is the Copernicus CMEMS product MULTIOBS GLO PHY REP 015 004-TDS at 15 m
depth. This product has a spatial resolution of 0.25° and a daily temporal resolution. It is derived from satellite
altimetry and model assimilation and includes both geostrophic and Ekman components. Using the surface velocity,
each particle within the circular region of a given sampling station was advected using a Runge-Kutta scheme of
order 4 from the day of the sampling backward in time. Different advective times were used, from 5 to 45 days.
Two types of diagnostics were carried out: Eulerian and purely Lagrangian diagnostics. These groups consist of
calculating properties that are integrated in time: at the sampling location (Eulerian) or along the trajectory of the
water parcel (Lagrangian). In this study, we only present diagnostics that are relevant to our area of study, such as
the Lagrangian and Eulerian chl-a, divergence, and vorticity. The Lagrangian chlorophyll, the average chl-a content
carried by the water parcel in the previous days, provides information on the recent primary productivity. The
Lagrangian divergence can be considered as a proxy of the upwelling (when negative) of downwelling (when
positive) experience by the water parcel in the previous days. This metric has been correlated with chlorophyll
(Hernández-Carrasco et al., 2018).
In the following, we will report diagnostics calculated using an advective time of 15 days. This value was chosen
as it showed the highest correlations between the chl-a concentrations and the abundance of micrometric particles
and macroscopic particles between 0-100m (Supplementary Fig. S8).
**2.2 Float data**
**2.2.1 Coverage and data collection**
For this study, a BGC-Argo float (WMO:6904139) was deployed at the equator during *RV Sonne* SO284 cruise
traversing the transect from 23°W to 7°W migrating from west to east during the period between July 2021 and
March 2022. The float was recovered during the PIRATA FR32 cruise. This float was equipped with several
physical and biogeochemical sensors to measure the pressure, temperature, salinity, chlorophyll, oxygen, and
particle backscattering coefficient (BBP) with a vertical resolution of 5 m. BGC-Argo float data were collected
through the International Argo Program and can be found at https://argo.ucsd.edu. Chl-a and BBP both present a
gap between the 1$^{st}$ and the 5$^{th}$ of January 2022.
**2.2.2 UVP measurements**
An Underwater Vision Profiler 6 (UVP6) was mounted on the BGC Argo float. This camera-based particle counter
sizes and counts marine particles (Kiko et al., 2022) covering a size range from 0.102 mm to 16.4 mm. The UVP
contributes to understanding sinking organic particles and carbon sequestration at global (Guidi et al., 2015) and
regional scales (Ramondenc et al., 2016). More information about calibration and data processing can be found in
Picheral et al. (2021). In total, our data set includes 86 profiles reaching at least 1000 m. Every 3 days, the BGC
Argo float reached 2000 m. For all parameters, we interpolated the data set with a vertical resolution of 10 m and
a temporal resolution of a day.
**2.2.3 Mixed layer depth calculation**
To determine the mixed layer depth, we use temperature profiles provided by the BGC-Argo float. Using the
definition outlined in De Boyer Montégut et al. (2004), the mixed layer was determined by identifying the depth at



which the temperature decreased by 0.2°C relative to the temperature at 10 m depth. The mixed layer depth in this
study reached a depth of 60 m.

**2.2.4 Particle abundance and carbon flux calculation**

Particle size abundances (number of particles per liter) for depth bins of 2.5 m along the water column were
obtained by the UVP. Particles were divided into two categories based on their size: Micrometric particles (MiP)
for particles ranging between 0.1-0.5 mm, and Macroscopic particles (MaP) ranging between 0.5-16 mm. The
carbon flux was obtained by integrating all size classes and therefore represents the total carbon flux. To calculate
the flux for a given size class, we used the relationship provided by Kriest (2002), linking the particle size to the
sinking speed and its carbon content. This relationship has been used in former studies using UVP observations
(Kiko et al., 2017). For each parameter, we interpolated the profiles in depth with a vertical resolution of 10 m and
a temporal resolution of a day.

**2.2.5 Determination of export events**

We determined periods of export events, using the anomalous carbon flux. We calculated the total mean particle
abundance and mean carbon flux along the water column from the interpolated fields for the deployment period.
The resulting mean profile was then subtracted from the individual particle abundance and carbon flux profiles,
yielding anomaly profiles. This helped us determine two different types of periods: periods with main export events
and periods where no or weak export occurred.

**2.2.6 Regime shift detection for surface export**

A sequential algorithm for regime shift detection (Rodionov, 2004) was applied to the MaP abundance for the first
200 m to identify accurately the beginning and the end of the carbon export events. This method identifies
discontinuities in a time series without prior assumptions of the timing of the regime shifts. The algorithm requires
a set of parameters to specify: the target significance level and the cutoff length. The target significance used here
is p=0.05. The cutoff length affects the time scale of the regime by removing regimes of shorter duration than the
reference value. In this study, the cutoff length was set to 9 days to cover at least 3 profiles. For more details, see
Rodionov (2004, 2006). We determined three masks, two corresponding to periods of export 'event 1', 'event 2',
and a period where no main export plume was observed, hereafter referred to as the 'outside-between' mask. It
should be noted that 'outside-between' refers to periods that do not belong to the two main export events.

**2.2.7 Morphological properties of detritus**

The data set consisted of 127,000 images. Each image underwent individual classification using the Ecotaxa
program with the support of machine learning classifiers (Picheral et al., 2017). This classification differentiates
between living and non-living organisms. The automatically classified images were then manually validated or
reclassified. To distinguish between different types of marine snow, we examined the morphological properties of
individual objects such as size (area, perimeter), shade intensity (mean/median gray level), shape (elongation), and
structural complexity (homogeneity or heterogeneity of gray levels). This was done using a principal component
analysis (Fig. S6) to summarize the morphological information into a few new variables, followed by k-means
clustering to separate different morphotypes of particles (Trudnowska et al. 2021). Using this method, we
distinguish between five types of marine snow, as this number was a good compromise between the continuum of



change in morphology and a need for simplicity. Concentration in numbers (numbers m$^{-3}$) was computed per 10 m
bins for each UVP6 profile.

**2.2.8 Flux attenuation and biological carbon pump efficiency**

The biological carbon pump (BCP) was computed following Engel et al. (2023), and Buesseler et al. (2020):
$BCP = E_{eff} \times T_{eff}$                                                                                     (1)
With E$_{eff}$, as the carbon export efficiency (E$_{eff}$)
$E_{eff} = \frac{F_{Z0}}{PP}$                                                                                      (2)
F$_{z0}$ is the carbon export flux out of the surface ocean layer, corresponding to 100 m. While PP is the amount of
CO2 fixed by primary production, both in mg m$^{-3}$ d$^{-1}$. Satellite-based net primary production (NPP) was
downloaded from the Ocean Productivity website (www.science.oregonstate.edu/ocean.productivity) using the
Vertically Generalized Production Model (VGPM)-Eppley.
T$_{eff}$ represents the carbon transfer efficiency:
$T_{eff} = \frac{F_Z}{F_{Z0}}$                                                                                      (3)
F$_Z$ is the flux at a particular depth and Z$_0$ is the reference depth (taken here as 100 m). T$_{eff}$ is related to the attenuation
of carbon flux with depth, over 0-1000 m, quantified by a Martin power law (Martin et al., 1987).
$F_Z = F_{Z0} \times \left(\frac{Z}{Z_0}\right)^{-b}$                                                              (4)
Z is the depth. The exponent b represents the attenuation with depth. An analogous equation was used to describe
the particle attenuation.
$n_Z = n_{Z0} \times \left(\frac{Z}{Z_0}\right)^{-b}$                                                              (5)
n$_Z$, n$_{Z0}$ are the concentrations of particles at depth Z or Z$_0$.

**3 Results**

**3.1 Satellite data analysis**

Throughout the float trajectory (Fig. S1), satellite observations disclosed the presence of relatively cold surface
waters during two distinct periods: August-October 2021 and December 2021 to February 2022 (Fig. 1a). The first
period aligned with the seasonal development and peak of the Atlantic cold tongue with minimum surface
temperatures around 23.8°C. The second period was from December to February with temperatures around 26°C.
The seasonal surface warming occurring between October and December featured values reaching 27.5°C, while
temperatures reached almost 29.5 during March and April (Fig. S2a). The temperatures were warmer than usual
compared to the climatology from 2012-2022, especially throughout the boreal summer of 2021 (Fig. S2a).





Before and during the cold tongue development in July to September 2021, bandpass-filtered SST anomalies
oscillated between -0.3°C and 0.3°C and showed westward propagation (Fig. 1b), suggesting the presence of TIWs.
A weaker TIW signal was observed during the second period when SST anomalies ranged between -0.1°C and
0.1°C.
Peaks of chl-a were observed during both low-temperature periods reaching about 0.4 mg m⁻³ on 10 September
2021 and 2 January 2022 (Fig. 1c, S2b). The surface chl-a concentration ranged between 0.1 and 0.4 mg m⁻³ along
the float trajectory. When comparing the chl-a to the climatology, a delay in both peaks was observed (Fig. S2b)
with a low peak during summer 2021 and a second high peak during winter 2022 for the float compared to the
climatology. The bandpass-filtered chl-a anomaly oscillated between -0.04 mg m⁻³ and 0.04 mg m⁻³ from August
to October (Fig. 1d). These anomalies seem to be anti-correlated with the SST anomalies (Fig. S2 c,d). However,
westward propagation of bandpass-filtered chl-a anomalies is less obvious than for SST. From December to March,
more pronounced chl-a anomalies were observed.

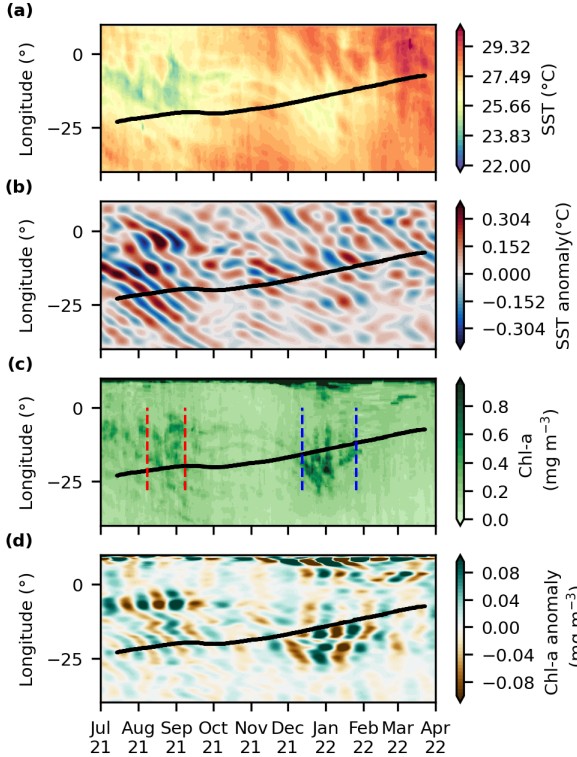


**Figure 1: Satellite-derived properties as a function of time (x-axis) and longitude (y-axis): (a) sea surface temperature (°C), (b) bandpass-filtered sea surface temperature anomaly, (c) surface chl-a concentration (mg m⁻³), and (d) bandpass-filtered chl-a anomaly along the equator from July 2021 to March 2022. The black line represents the float trajectory from west to east. The blue and red lines determine the beginning and the end of the first and the second export event, respectively.**



### 3.2 Float data analysis

### 3.2.1 Physical parameters (Temperature, Salinity, mixed layer)

During August, the mixed layer depth (MLD) was at about 42 m, salinity integrated over the first 100 m showed a maxima of 36 PSU (Fig. 2a, S3a) while temperature (averaged over the top 100 m) showed a first minimum of 21°C (Fig. 2b). Average 100-m temperature increased with the progressive deepening of the ML from September to January. MLD reached a maximum of 60 m in October. Salinity was high during the same period with a maximum of 36 psu in November (Fig. 2a). A second 100-me temperature minimum of 18.8°C was also recorded in January coinciding with the shoaling of the Deep Chlorophyll Maximum (DCM) and the presence of a salinity minimum in the top 100 m (Fig 2a,b, S3a,b). A subsurface maximum was determined for the salinity between 50-100 m (Fig. 2a,b). The thermocline, represented by the 20°C isotherm (black line, Fig. 2,e) was unusually deep during summer (80 m) compared to the Argo climatology and reached a maximum of 100 m in October. It was also unusually shallow (50 m) in January 2022.

### 3.2.2 Biogeochemical parameters (chl-a, BBP, oxygen)

The chl-a concentration reported by the BGC-Argo float in the first 100 m, varied between 0 and 0.5 mg m$^{-3}$. Peaks reached 0.45-5 mg m$^{-3}$ values on September 18, November 3, and 8 December (Fig. 2d, S3a). Elevated chl-a concentrations were observed at 70 m depth, corresponding to the depth of the deep chl-a maximum. These values varied between 0.28 and 0.8 mg m$^{-3}$. Both satellite and float chl-a data show the presence of two blooms (Fig. S3a). However, float data presented a more variable chl-a concentration compared to the satellite. This can be attributed to the low resolution of satellite images compared to the float and the interpolation methods applied to ensure a gap-free time series. BBP POC, calculated using a BBP-to-carbon relationship (Koestner et al., 2022), followed the same pattern as chl-a and small particles were concentrated in the first 90 m (Fig. 2e). Periods of high chl-a were correlated with an increase in the BBP POC. Oxygen concentrations reached values around 189 µmol kg$^{-1}$ (Fig. 2c) in the mixed layer. Concentrations decreased with depth, with values below 134 µmol kg$^{-1}$ below 100 m.

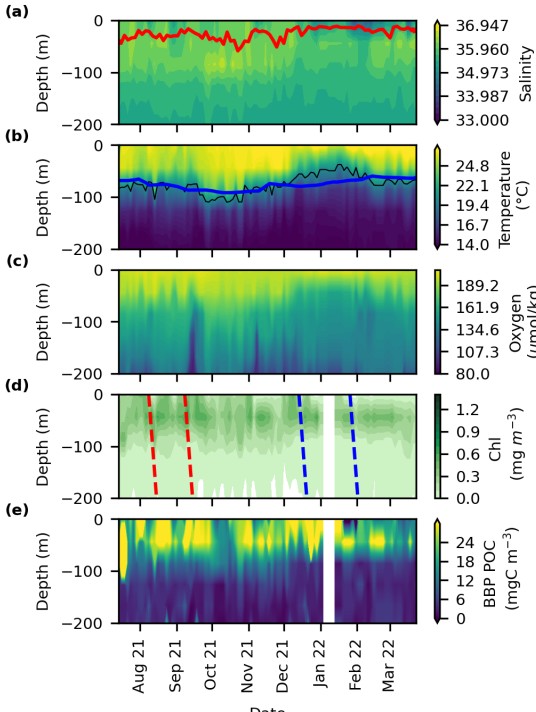

**Figure 2: Time–depth profiles determined from the BGC-Argo float for (a) salinity, (b) temperature (°C), (c) oxygen**
**(μmol kg⁻¹), (d) chl-a (mg m⁻³) (e) and BBP POC (mgC m⁻³). The red line in (a) represents the mixed layer depth defined**
**as a decrease of 0.2°C relative to temperature at 10 m depth. The black line in (b) represents the 20°C isotherm depth**
**which is a well-known proxy for the thermocline in the tropics. The blue line in (b) is the average depth of the 20°C**
**isotherm from the Argo climatology (2012 to 2022). The blue and red dashed lines in (d) determine the beginning and**
**the end of the first and the second export event, respectively.**
**3.3 Carbon flux dynamics**
**3.3.1 Surface flux and particle abundance along the trajectory (0-100m)**
The increase in surface chl-a in the first 100 m was linked to an increase in surface carbon flux and MiP abundance
(particles between 0.1-0.5 mm) (Fig. S3). Both were significantly correlated with in situ chl-a ($r^2$=0.4 and 0.3,
respectively, p-value<0.01). No significant correlation was found between surface chl-a and MaP abundance
(particles>0.5 mm). The highest integrated MiP abundance in the surface layer was recorded on the 18[th] of August
2021 with values reaching 316 particles L⁻¹ (Fig. 3a,c, S3d). This also coincided with the highest MaP abundance
with around 5 particles L⁻¹. Simultaneously with the surface chl-a peak, on 3 November 2021 a peak of MiP with
348 particles L⁻¹ was also observed (Fig S3d), while carbon flux increased after a 15 day delay reaching 250 mg C
m⁻² day⁻¹ (Fig. S3f). The peak of chl-a, in December 2021 caused an increase in carbon flux, MiP, and MaP
abundance.
**3.3.2 Flux and particle abundance pattern along the water column (averaged profile)**
Throughout the float trajectory, the UVP6 data showcased high variability in MiP and MaP abundance, and carbon
flux in the upper 100 m, with a dominance of small particles compared to big particles (Fig. 4a, S3). MiP and MaP
increased in the surface layer peaking at 30-40 m, respectively. The maximum carbon flux, 104.1 ± 61.5 mgC m⁻³



day$^{-1}$, coincided with the MaP's maximum. After the surface layer peaks, MiP abundance and carbon flux declined
rapidly until 1000 m, reaching 22.5 ± 1.3 particles L$^{-1}$ and 13.2 ± 2.9 mgC m$^{-3}$ day$^{-1}$, while MaP's abundance
decreased rapidly until 200 m with 0.3± 0.2 particles L$^{-1}$. Flux and abundance declined further with depth. The
carbon flux was dominated by MiP abundance and followed its pattern.

### 3.3.3 Evaluation of export events

To investigate two settling plumes depicted in Figure 3d, four spaced lines were drawn on the MaP abundance with
a slope of 30 m day$^{-1}$, as suggested by Stemmann, Jackson, & Gorsky, (2004) using a model for particle size
distribution. The periods of surface production and export were determined using the Rodionov algorithm (Fig.
S4). The first export event, "Event 1", started at the surface on the 8th of August and lasted until the 8th of
September 2021, while the second event, "Event 2", occurred from the 13th of December 2021 to the 26th of
January 2022. Both events lasted one month (Fig. 3f). These events are easily discernible on the MaP abundance
and carbon flux plots as two plumes that reach 2000 m depth (Fig. 3c-f) while they are less visible in the MiP
pattern.

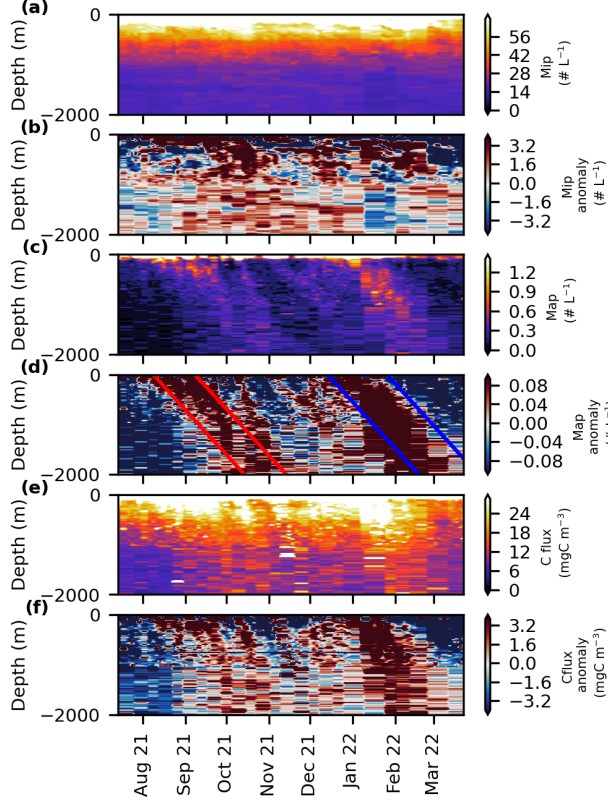

**Figure 3: Time series of (a) MiP abundance (# L$^{-1}$), (b) MiP anomaly, (c) MaP abundance (# L$^{-1}$), (d) MaP anomaly, (e) carbon flux (mgC m$^{-3}$ day$^{-1}$), and (f) carbon flux anomaly. The blue and red lines determine the beginning and the end of the first and the second export event, respectively.**





Interestingly, the carbon flux profiles for events 1 and 2, and the 'outside-between' mask showed the same
attenuation of the mean carbon flux along the water column (Fig. 4a,b). Flux at 30 m depth reached 168, 139, and
83 mg C m$^{-3}$ day$^{-1}$ for event 2, 1 and outside-between mask, respectively (Fig. 4a). The flux then decreased with
depth during all periods. An intermediate particle maximum was observed for 'outside-between' and event 2
between 300-500 m (Fig. S5). The 'outside-between' mask showed the lowest carbon flux along the water column
compared to the two export events. Flux during event 2 was the highest from 0-60 m and then from 120 m to 2000
m (Fig. 4a-c). Post hoc Tukey tests showed a significant difference between the outside-between mask and the two
main export events for most of the layers of the water column (Fig. 4), while there was rarely a difference between
events 1 and 2.

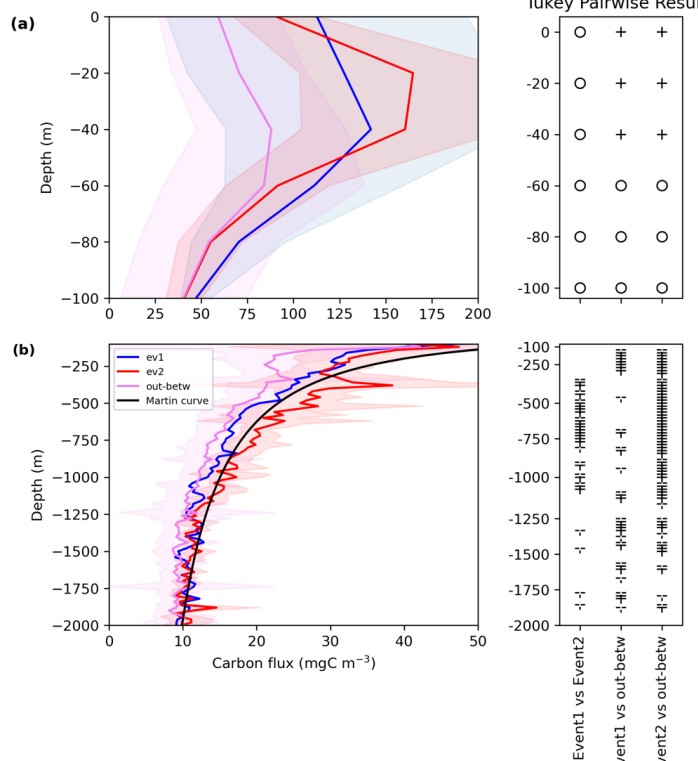


**Figure 4: Averaged carbon flux profiles (mg C m$^{-3}$) along the plumes during event 1 (blue), event 2 (red), and the outside-between mask (purple) (a) from 0-100 m, (b) from 200-2000 m. The shading represents the standard deviation. The Tuckey pairwise results were conducted for each depth. Plus signs indicate a significant difference, and blank space or empty circles indicate a non-significant difference. The black line represents the Martin curve of event 2 calculated using Eq. 4 with b=-0.6 and F$_{z0}$= 100m .**

**3.3.4 Flux attenuation and export efficiency**
We parameterize the strength of the BCP pump using the export efficiency calculated at 100 m. Export efficiency
ranged between 6-7% of the NPP (from satellite data estimates) exiting the 0-100 m layer. The attenuation rate of
carbon flux was determined using a power law regression fit. The b values ranged between - 0.4 and - 0.6. The best



transfer efficiency was found during event 2, where 40% of the flux at 100m reaches 1000m, followed by the
between-outside mask and event 1 (31% and 29%, respectively; Table 1).
**Table 1: Parameters characterizing the biological carbon pump efficiency calculated in the plumes**

|                 | Outside-between mask | event 1 | event 2 |
|-----------------|----------------------|---------|---------|
| $E_{eff}$       | 7%                   | 7%      | 6%      |
| $T_{eff}$       | 31%                  | 29%     | 40%     |
| b               | -0.48                | -0.53   | -0.6    |

**3.4 Particle composition**
**3.4.1 Morphotypes of marine snow and composition of the different events**
The k-means clustering applied to the PCA coordinates helped us to distinguish between five marine snow
morphotypes illustrated in Figure 5. Type 1 consisted of big and dense objects (Big Dense Particles, BDP) with an
Equivalent Spherical Diameter (ESD)>0.8mm. Type 2 comprised elongated objects (Fiber particles: FP), and type
3 consisted of big, bright, and porous objects (Big Porous particles: BPP). Type 4 was mainly formed of dense,
small, and circular objects (Small Dense particles: SDP) and type 5 consisted of light grey, small, and porous
objects (Small Porous particles: SPP). These five morphotypes were then used to characterize the distribution and
composition of marine snow. It should be noted that the terms "porous" and "dense" refer to brightness, with
"porous" indicating greater light transmission.



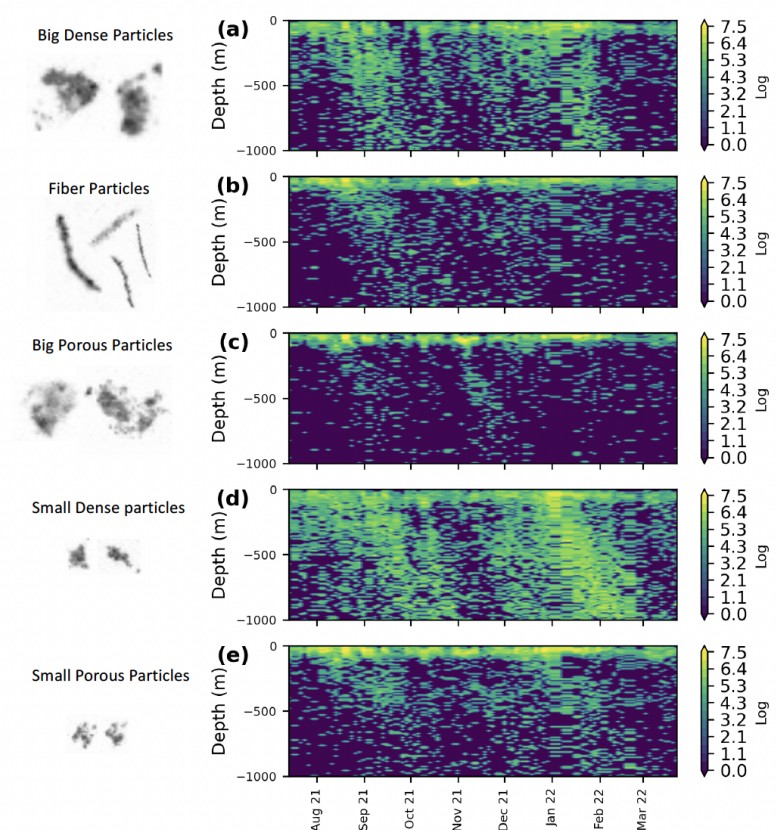


**Figure 5: Time series of the logarithmic concentration of (a) Big Dense, (b) Fiber, (c) Big Porous, (d) Small Dense, and (e) Small Porous morphotypes.**

The different detritus morphotypes showed high concentrations of particles in the surface, especially during the export events (Fig. 5, S7). They shared similar temporal dynamics primarily in the surface layer: FP, BPP, BDP, SDP, and SPP decreased exponentially between 0 and 150 m. While FB, BPP and SPP decreased slowly throughout the water column in the mesopelagic layers, BDP and SDP increased gradually between 400-600 m and then decreased again (Fig. S7). BDP and SDP presented two discernable plumes during the two delineated export events, reaching 2000 m with significant concentrations all along the plumes (Fig. 6a,d, S6). Other morphotypes such as FP and SPP were sometimes present in deeper layers but with low concentrations. In the deeper mesopelagic, only SDP showed no decrease with depth (Fig. S7).

SPP were most abundant across the different periods (Fig. S7). For the outside-between mask, these particles constituted 28% within the 0-100 m range. In events 1 and 2, their presence increased up to 50-60% within the 0-50 m range (Fig. S7). However, this percentage notably declined between 50-100 m for events 1 and 2 with values dropping to 25% below 100 m. It further decreased between 100-150 m outside of these events, 18% below 150 m (Fig. S7). BPP and FP also exhibited high concentrations within the upper 100 m where they were primarily located.



Even though BDP and SDP were the least preponderant classes in the first 100 m, they dominated deeper layers
compared to clusters BPP, SPP, and FP which were found almost exclusively in the surface layer. Their proportions
were more than 20% for events 1 and 2 and 15% outside of these export events in the first 100 m. Concentrations
of small, dark, and compact marine snow decreased until 200 m and then increased until 1000 m, while big, dark,
and compact particles decreased until 400-450 m for all masks and increased afterward (Fig. S7). Small compact
particles showed a mean proportion of more than 40% for both events and 28% for the period outside of those (Fig
S7). Throughout the observation period, the vertical attenuation of dark, compact, and small particles was the
lowest of all marine snow categories with visibly more dark dense morphotypes in deep waters for all periods,
while the highest vertical attenuation amongst all marine snow types was observed for the BPP cluster.
**3.5 Lagrangian diagnostics**
To determine the optimal advective time scale for the different particle sizes, we correlated both MiP and MaP
abundance in the first 100 m with the Lagrangian chl-a (chlorophyll in the moving water mass) for the different
advective times (from 0-45 days). The highest correlation was determined for t=15 days (Fig. S8). Chl-a and BBP
POC were positively correlated with MiP in the first 100 m (Fig. S9). While MaP showed a low correlation with
chl-a. As for the different morphotypes, FP, SDP, and SPP were significantly correlated with chl-a while no
significant correlation was observed with BDP and BPP. POC flux, MiP, and MaP abundances were also
significantly correlated with the Lagrangian vorticity and divergence for an advective time scale of 15 days.
**4. Discussion**
**4.1 Cold tongue and TIW related equatorial upwelling dynamics**
The equatorial Atlantic follows a pronounced seasonal cycle in upwelling activity, forced by the seasonal winds
and the meridional migration of the intertropical convergence zone (Brandt et al., 2023), which translates into a
respective cycle of productivity (Grodsky et al. 2008) and a slight seasonality in carbon export (Fischer et al., 2000;
Wefer & Fischer, 1993).
Here, we use a combination of satellite data analysis and in situ biogeochemical and image-based measurements
from a BGC-Argo / UVP float to further our understanding of equatorial Atlantic biological pump dynamics. We
observed relatively cool SST between August and October corresponding to the occurrence of the Atlantic Cold
Tongue (ACT; Brandt et al., 2011), and during boreal winter in December and February corresponding to a
secondary cooling period as was reported by Jouanno et al. (2011) and Okumura & Xie (2006). Seasonal cooling
events are largely linked to diapycnal heat flux out of the mixed layer into the deeper ocean (Hummels et al., 2013).
This heat flux is due to the enhancement of the vertical shear driven by the strength and the direction of the surface
current (Jouanno et al., 2011) and the eastward Equatorial Undercurrent (EUC) at the thermocline level (Hummels
et al., 2013). The equatorial Atlantic was warmer than usual at the surface throughout the boreal summer of 2021.
This was the result of the occurrence of a strong Atlantic Niño driven by wind and equatorial wave forcings (Lee
et al., 2023; Song et al., 2023; Tuchen et al., 2024). The physical processes controlling the downward heat flux out
of the mixed layer also control the upward supply of nitrate to the euphotic layer (Radenac et al., 2020). The
equatorial Atlantic is a nitrate-limited upwelling regime (Grodsky et al., 2008; Moore et al., 2013) and modeling
studies showed that in the equatorial Atlantic, the seasonal variations in chl-a are closely linked to the seasonal
variability of the nitrate input via upwelling and mixing (Loukos & Mémery, 1999; Radenac et al., 2020). The chl-



a blooms that are normally found in the equatorial Atlantic are linked to the upwelling of nitrate-rich thermocline
waters during these periods and the diffusive flux of nitrate into the mixed layer through mixing (Longhurst, 1993;
Radenac et al., 2020). Likewise, we found that chl-a concentration followed a pronounced semiannual cycle with
peaks in boreal summer and winter, as also described by Grodsky et al. (2008) and Brandt et al. (2023). During the
first peak, from August to October, the thermocline was relatively deep compared to climatology (as a consequence
of the presence of the Atlantic Niño), and chl-a levels were likewise relatively low. A shallower nitracline together
with a shallower EUC (Tuchen et al., 2024) during the second peak in boreal winter might have favored the growth
of the phytoplankton assemblage showing anomalously high chl-a levels. These variations with respect to the
climatological cycle are in agreement with what was proposed by Grodsky et al. (2008) that the interannual
variability of the secondary bloom in boreal winter is as large as those of the primary bloom in boreal summer,
even though its climatological expression is weaker.
Another process affecting local productivity at the equator is intraseasonal TIWs with a 20-50 days period range
as indicated by the bandpass-filtered SST and chl-a anomalies. In this study, elevated primary production was
located between 10°W-25°W in a region affected by TIWs. Their occurrence strongly suggests that TIWs might
influence the biogeochemistry of the equatorial upper-ocean system. On the one hand, TIWs are associated with
meridional currents at the equator modulating the boundary of the Atlantic cold tongue eventually resulting in local
variations of SST and chl-a. On the other hand, TIWs are associated with phases of enhanced mixing (Foltz et al.,
2020; Heukamp et al., 2022; Inoue et al., 2019; Moum et al., 2009) or front generation (Warner et al., 2018) leading
to upward nutrient supply. It has been suggested that TIWs could enhance upper-ocean fertilization by promoting
local nitrate upwelling alleviating the nitrate depletion which usually affects this region (Radenac et al., 2020;
Sherman et al., 2022). Enhanced chlorophyll concentration has been associated with TIWs suggesting that TIWs
drive intraseasonal chl-a variability (Grodsky et al., 2008; Menkes et al., 2002; Shi & Wang, 2021). Pronounced
positive and negative anomalies in bandpass-filtered chl-a data were anti-correlated with anomalies in bandpass-
filtered positive SST anomalies. The SST anomalies were moderate during the secondary bloom in boreal winter,
accompanied by a shallower thermocline. However, during that period, a pronounced chl-a bloom was observed
together with the largest bandpass-filtered chl-a anomalies.
In brief, the development of the cold tongue during boreal summer, the secondary cooling during boreal winter,
and the presence of TIWs in the equatorial Atlantic exert major controls on the surface ocean hydrographic
characteristics and biogeochemistry on intraseasonal to seasonal time scales. We suggest that the combination of
seasonal thermocline upwelling and TIWs was responsible for the observed enhanced chl-a signals indicating
enhanced variability of primary productivity. Therefore, we can examine their impact on particulate matter build
up and export.
**4.2 Upwelling events translate into size-differentiated enhanced export from the mixed layer**
The timing of the two upwelling events which lead to chl-a accumulation is consistent with the objective detection
of two peaks in the MaP concentration. Both MiP and MaP in top 100 m are correlated to the in situ chl-a biomass
suggesting that the primary producers provided the elemental particles for the two size classes of marine snow
aggregates. Stronger correlation with MiP than with MaP may indicate that MaP are formed with a delay through
the transformation of MiP by aggregation. This is also supported by Lagrangian chl-a which is more correlated



with MiP and MaP (for the same advective time scale of 10 to 15 days) than with concomitant in situ chl-a biomass.
This time scale is consistent with particle aggregation by coagulation of phytoplankton cells followed by the export
of aggregates (Burd & Jackson, 2009; Jackson, 1990). Correlation between Lagrangian chl-a and MiP emphasizes
that MiP are also built up with time.
More comprehensive understanding of pelagic functioning can arise from the identification of marine snow
morphotypes (Trudnowska et al., 2021). Fiber Particles (FP) , Small Dense particles (SDP), and Small Porous
Particles (SPP) in the epipelagial were significantly correlated with chl-a while no significant correlation was
observed for BDP and BPP. This means that both fiber and porous aggregates might be of phytoplanktonic origin.
Elongated or porous particles in the surface layer, for example, can result from phytoplankton colonies such as
diatom chains or *Trichodesmium* colonies (Dupouy et al., 2018; Villareal et al., 2011) (Fig. S10). These diatoms
were mostly detected during event 1 in our study, their presence increases in conditions of high export systems
(Henson et al., 2019). Porous aggregates might be associated with the accumulation of phytoplankton biomass.
When the bloom is massive enough to enhance aggregation, small porous aggregates are precursors of bigger ones.
As for dense particles, they could be potential fecal pellets produced by zooplankton's feeding as generally found
in other studies (Stemmann & Boss, 2012; Trudnowska et al., 2021).
All morphotypes, particle size classes and POC flux were significantly correlated with Lagrangian vorticity and
divergence highlighting that physical dynamics of the upper ocean (such as up- and downwelling) leading to
primary production were the primary control in particle production and transformations. In their paper, Siegel et
al. (2024) found that turbulence levels close to the surface tend to favor smaller particle sizes and increase
fragmentation while turbulence near the base of the mixed layer encourages coagulation and the formation of larger
particles.
POC production is associated with phytoplankton production which ultimately influences the export flux. To
investigate the ratio of POC flux leaving the euphotic zone, we calculated the export efficiency ratio at 100 m
during and outside of the events. The export efficiency (e-ratio) was 6-7% and fell within the global average e-ratio
range (Bam et al., 2023 and references within). These values suggest that strong remineralization occurs in surface
waters, aligning with existing literature (Clements et al., 2022). The same e-ratio during and outside of events
highlights the stability of the export efficiency of the equatorial system. One of the hypotheses explaining this
stability is the distribution and contribution of the morphotypes during and outside of export events. During and
before export events, all five morphotypes were detected, with proportions varying with depth. This suggests that
within our observation period, the equatorial region, during or before export events, possesses a similar
phytoplanktonic bloom behavior leading to the same marine snow morphotypes which might explain the similar
behavior of the biological pump. This is in contrast with what was observed for the Arctic system, where two
successive blooms of different nature occur and are associated with different morphotypes. The first bloom was an
ice edge bloom and was dominated by diatoms, while the second was ice free and was associated with the presence
of *Phaeocystis* leading to agglomerated morphotypes and their slow settling compared to the first bloom
(Trudnowska et al., 2021).
Another hypothesis for the stability of the system might be related to the tight coupling between primary production
and export. In this study, the lag between PP and particle production was estimated to be 10-15 days, corresponding



to a similarly short lag determined by Henson et al. (2015) usually found in upwelling regions. This lag increases
with the increase of seasonality and also affects the seasonality of the $e_{eff}$ (Henson et al., 2015). In our case, the
same $e_{eff}$ highlights the low seasonality of the carbon pump in the equatorial system: producers and grazers are
tightly coupled due to the low seasonality in PP and export (Owens et al., 2015). This coupling might be due to the
combination of euphotic-zone irradiance and the supply of nutrients: strong light penetration combined with the
energetic intraseasonal variability of the system bringing nutrients to the surface (Menkes et al., 2002), allows
producers to be present all year long in the surface layer. Further studies on the dynamics and composition of
detrital particles in bloom situations, in combination with planktonic measurements, are necessary to understand
surface dynamics of particle formation and export.
**4.3 Deep particle sequestration is driven by compact particles**
Particle production within the upper 100 m led to the formation of sinking plumes reaching down to 2000 m of
depth. Although particle concentrations were higher during export events, the vertical carbon flux, within and
outside the plumes followed the general asymptotic shape characteristic of particle flux observations, with rapid
attenuation in the surface layer transitioning to a more gradual decrease in the bathypelagic layer. This was also
true for the MiP and MaP abundances. More likely, the observed general decrease in small and big particles is
driven by biological processes such as degradation and aggregation. Yool et al. (2013), using a biogeochemical
model, attributed the flux of particles at deep depths of the ocean to MaPs. However, Kiko et al. (2017) found an
abundance of MiPs in the bathypelagic zone that can be observed down to the sea floor. They suggest that shedding
and other disaggregation processes might result in a more effective export of particulate matter, both actively and
passively. We suggest that their presence in the meso- and bathypelagic layer highlights both their important roles
in contributing to the flux. The difference in flux amplitude inside versus outside of the plumes, along with the
higher particle concentration within the plumes, suggests a seasonal pulse in flux to the deep sea, as previously
described by Beaulieu (2002). This rapid and deep flux is mainly associated with bloom events, consistent with
earlier observations of flux events reaching depths of up to 4000 m (Beaulieu, 2002; Kiko et al., 2017; Lampitt et
al., 1993).
We combined the quantitative analysis of particle mass distribution with particle image analysis to investigate the
nature of particles exported to deeper layers. In this study, SDP were more deeply exported compared to other
morphotypes, indicating that most of the MaP abundance found at depth is dominated by small dense particles.
This trend toward more circular and less elongated aggregates with increasing depth confirms prior research
(Drago, 2023; Trudnowska et al., 2021; Accardo et al., submitted). SDP vertical profiles also showed a particle
maximum between 450 and 800 m as found by Kiko et al. (2017) and Siegel et al. (2024), unlike the rest of the
morphotypes which attenuated with depth. The observed increase in small particles is more likely driven by diel
vertical migrations of zooplankton that actively exports organic material to depth (Hidaka et al., 2001; Turner,
2015). This can be confirmed by the increase of zooplankton for all periods between 300-600 m (Fig. S11)
coinciding with the depth of increase of dense particles and the upper limit of zooplankton migration depth
extracted from shipboard ADCP data (shown in Fig. S11) and the range mentioned in Kiko et al., (2017) and
Bianchi & Mislan, (2016). Kiko et al. (2017) also found a particle maximum between 300 and 600 m in the
equatorial Atlantic and attributed it to migrating zooplankton. Food ingested near the surface is carried downward
in the guts of migrating zooplankton to be egested, eaten by consumers of zooplankton, or metabolized at depth



(Packard & Gómez, 2013). Model studies suggest that zooplankton diel vertical migration might account for 10-
30% of the total vertical flux of carbon downward from epipelagic layers (Bianchi et al., 2013), enhancing the
efficiency of carbon export (Gorgues et al., 2019) through the generation of fecal pellets which can be incorporated
in marine snow. Kiko et al. (2020) found that gut flux and mortality might make up about 30-40% of particulate
matter supply to the 300-600 m depth layer in the eastern tropical North Atlantic and that the amount of carbon
supplied via these mechanisms could suffice to generate a flux and particle increase.
Because our study includes periods of both high and low export, we aimed to assess the flux attenuation rate by
calculating the transfer efficiency for different periods. The transfer efficiency was estimated using flux at depths
of 100 m and 1000 m. Although our export reached depths of 2000 m, we selected the 1000 m layer due to the
relatively small reduction in POC flux with increasing depth, as noted by Francois et al. (2002). $T_{eff}$ values obtained
in our study indicate a high efficiency of the biological pump with up to 40% of the organic material exported at
100 m also reaching 1000 m, regardless of the conditions. This means that despite seasonal variation in primary
production and carbon flux during and outside of export events, the biological pump exhibits a consistent response
in the equatorial region, rendering it a predictable system. The b values calculated (around 0.5) show a low
attenuation rate of the computed POC flux, suggesting that part of the particulate matter exported at the equator
and undergoes little further remineralization at mesopelagic depths (Henson et al., 2015; Omand et al., 2015).
Global studies showcased the seasonal and regional variability in the exponent b and showed values around 0.7 in
Guidi et al, (2015) and 0.6 in Henson et al., (2012) for the tropical equatorial region. The consistent low e-ratio
associated with high $T_{eff}$ aligns with the pattern proposed by Guidi et al, (2015) and Henson et al., (2012). This
means that deeper particle injection and rapid sinking result in longer carbon sequestration as the time a given water
parcel needs to travel from the ocean interior to the surface increases with depth.
For the first time, we characterized the distribution of particles within an export plume, and offered a morphological
description of exported particles using in situ imaging. The comparisons made with previous studies for the e-ratio
and the transfer efficiency show that opting for the plume method yields more accurate results and a more
comprehensive understanding of the fate of particles along their progression into deeper layers (Table S1).





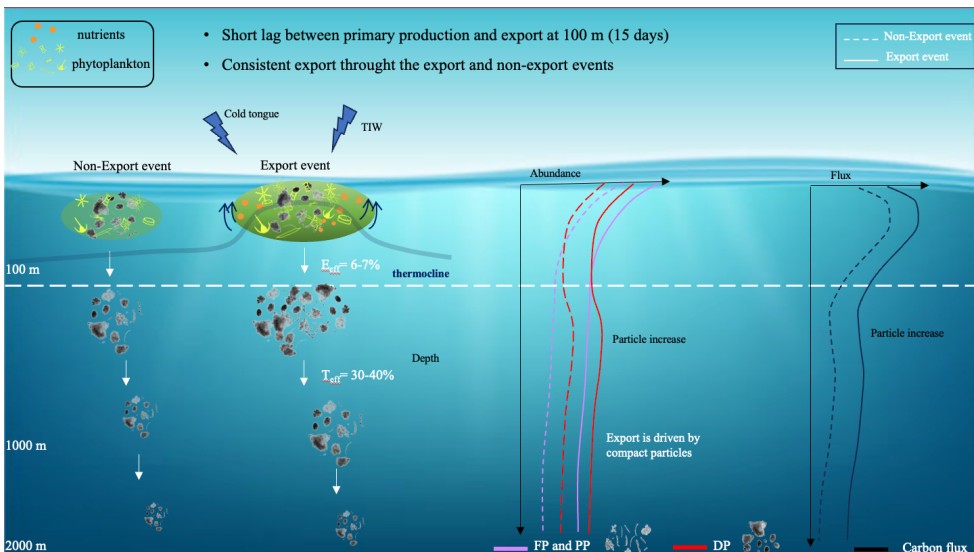


**Figure 6: Illustrative example of the particle export system in the Atlantic equatorial region during export and non-export events. FP: fiber particles, PP: porous particles, DP: dense particles. Teff: Transfer efficiency, Eeff: Export efficiency, TIW: Tropical instability Wave.**

### 5 Conclusion

The integration of the UVP on an Argo float has allowed us to study the temporal variability and the dynamics of the BCP. Our study follows the particle dynamics along the water column in the equatorial Atlantic region between July 2021 and March 2022 including the strong 2021 Atlantic Niño event, using the plume method and a novel BGC-Argo/UVP6 dataset. Ocean dynamics in the equatorial system exhibit a seasonal cycle with a decrease in temperature during boreal summer and winter leading to the presence of two distinct blooms. These blooms are characterized by significant export events reaching depths of 2000 m. The production and export of carbon during that year was dampened because of the strong Atlantic Niño event during boreal summer 2021. Detritus were classified into five distinct morphotypes based on morphological variables. In surface waters, marine snow is dominated by porous aggregates and fibers while deeper layers primarily receive big and small dense particles during export events. Unlike most of the morphotypes decreasing with depth, dense particles show an increase between 300-600 m. Zooplankton diel vertical migration might play a role in the generation of a particle maximum at intermediate layers consisting of the small dense cluster. The equatorial region acts as a stable export system throughout all periods observed, with an export efficiency steadily ranging between 6-7% probably due to the short lag between the primary production and the export and the same morphotype composition along the year. Regardless of the initial conditions, 30-40% of the flux at 100 m is exported to 1000 m. Such consistency highlights the equilibrium inherent in the equatorial region's carbon dynamics along the float trajectory during this special event, providing further context to the observed patterns of carbon and particle export. Moreover, it underscores the necessity for additional observations to ascertain whether the system is truly stable over the long term. This study contributes to a deeper understanding of the intricacies of carbon cycling in equatorial waters using autonomous vehicle-derived estimates of particle fluxes. By elucidating the role of export events and different particle morphotypes, we underscore the significance of these factors in shaping the equatorial biological pump.



The successful combination of the UVP6 with other float sensors and the development of a continuous monitoring
strategy will provide insights that were previously unattainable with sparse and temporally limited shipboard and
moored sediment trap observations.
**6 Code availability**
The codes used post-data treatment are available upon request to the lead author.
**7 Data availability**
The    sea    surface    temperature    data    is    available    on    the    NOAA    website    at:
https://psl.noaa.gov/data/gridded/data.noaa.oisst.v2.highres.htm. Float data are available at https://argo.ucsd.edu.
Data used in this manuscript for the carbon flux and particle concentrations are available online using the following
DOI: 10.5281/zenodo.14007570. Further data can be made available by the authors upon request.
**8 Author contribution**
JH, RK, and LS developed the study's concept. RK, LS, JH, FPT, PB, and AB contributed to data acquisition; RK,
LS, JH, PB, AA, AB, and FPT contributed substantially to the data analysis; JH wrote the initial version of the
paper. All authors contributed substantially to drafting the manuscript; All authors approved the final submitted
manuscript.
**9 Competing interests**
The authors declare that they have no conflict of interest.
**9 Acknowledgments**
The study was supported by EU H2020 under grant agreement 817578 TRIATLAS project. RK acknowledges
support via a Make Our Planet Great Again grant from the French National Research Agency (ANR) within the
Programme d'Investissements d'Avenir #ANR-19-MPGA-0012 and funding from the Heisenberg Programme of
the German Science Foundation #KI 1387/5-1. We thank the crews, scientists, and technicians involved in the
deployment and recovery of the Argo float during RV *Sonne* cruise SO284 and PIRATA FR32 cruise with RV
*Thalassa*.

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
