# Peer review of "Marine snow surface production and bathypelagic export at the"

_EGUsphere, 2024_

## Author Comment (AC1)

**Marine snow surface production and bathypelagic export at the Equatorial Atlantic from an imaging float**

Joelle Habib, Lars Stemmann, Alexandre Accardo, Alberto Baudena, Franz Philip Tuchen, Peter Brandt, and Rainer Kiko

**Responses to the Reviewers' Comments**

Answers to reviewers' comments are reported point by point. The questions and comments of the reviewers are in blue, the answers in black, and the modifications that we made in the revised manuscript in red.

**Responses to the comments of the anonymous Reviewer 1**

First, we would like to warmly thank Reviewer 1 for their relevant and constructive comments, which helped to improve the manuscript.

*The study provides a comprehensive data set, observing and describing two export events in the equatorial Atlantic and key factors that may drive the observed carbon flux. They present details on particle properties such as their size and density. I think that is a valuable case study of how recent developments in underwater imaging technology can be used to gain detailed insights into carbon flux mechanisms. The data analysis and method description are clear and thorough.*

**Reply:** We appreciate the positive assessment of Reviewer 1.

*L44–56: Mineral ballasting could be mentioned as an additional factor*

**Reply:** This will be added as following in the revised manuscript.

*Mineral ballasting, through the association of marine snow with dense inorganic materials such as calcium carbonate, lithogenic or biogenic silica, can also significantly enhance sinking velocities and control the carbon export efficiency (Armstrong et al., 2002; Klaas and Archer, 2002).*

*L315: "best transfer efficiency" – rephrase to "highest transfer efficiency" to avoid a misleading qualitative assessment*

**Reply:** This will be rephrased as suggested in the revised manuscript.

*Section 3.4.1. I found this section difficult to read as five acronyms were introduced at once. In many points the authors make, there is a pattern by either size or by packaging. I would suggest spelling this out by describing the categories as either small or large, and either densely packed, loosely packed, or fiber. Also, I think that the word "porous" is misleading; I would suggest "dense" and "loose" or "dark" and "light", based on the image property. Example: "They shared similar temporal dynamics primarily in the surface layer: all types decreased exponentially between 0 and 150 m. While fibers and loose/light particles decreased slowly throughout the water column in the mesopelagic layers, dense/dark particles increased gradually between 400-600 m."*

**Reply:** We thank the reviewer for this thoughtful comment. While we understand the concern about introducing multiple acronyms at once, we believe that the classification is important for distinguishing the five morphotypes, which reflect distinct features relevant to our analysis. We recognize, however, that the section may benefit from improved clarity. Rather than replacing the acronyms entirely, we prefer to retain them but will ensure that each is introduced with accompanying descriptive terms (e.g., small/large, dense/loose, or fiber-like) to improve readability.

Regarding the use of the term "porous," we acknowledge that interpretation can vary. However, we use this term to reflect specific textural characteristics observed in the images, and we are cautious about replacing it with terms like "loose", which may carry a different meaning. We note that Reviewer 2 may provide additional perspective on this point, and we prefer to consider all feedback collectively before making substantial changes to terminology or structure. We are open to further adjustments depending on the broader consensus across reviews.

*The k-means clustering applied to the PCA coordinates helped us to distinguish between five marine snow morphotypes illustrated in Figure 5. Type 1 consisted of large, compact objects with an Equivalent Spherical Diameter (ESD)>0.8mm referred to as Big Dense Particles (BDP). Type 2 comprised elongated, thread-like objects termed Fiber particles (FP), and type 3 consisted of large bright, and porous objects referred to as Big Porous particles (BPP). Type 4 was mainly formed of dense, small, and circular objects: Small Dense particles (SDP), and type 5 consisted of small, bright, and porous objects: Small Porous particles (SPP). These five morphotypes were then used to characterize the distribution and composition of marine snow. It should be noted that the terms "porous" and "dense" refer to brightness, with "porous" indicating greater light transmission while "dense" denotes lower light transmission.*

*The morphotypes identified here are different than the clusters identified by Trudnowska et al (which are cited here). Discussing why different categories were identified in the context of this ecosystem would add value to the paper.*

**Reply:** We agree that addressing the differences between our morphotype categories and the clusters identified by Trudnowska et al. (2021) would strengthen the manuscript. Trudnowska et al. categorized marine snow into five broad morphotypes (dark, elongated, flake, fluffy, and agglomerated), optimized for the Arctic environment, particularly to capture the high morphological variability across bloom phases and changing phytoplankton communities in marginal ice zones. Their emphasis was on temporal structural changes concerning Arctic bloom dynamics. In contrast, our study, conducted in the equatorial Atlantic, defines five morphotypes that we interpret as reflecting morphodynamic features specific to this region's ecological and physical context. We have now added a brief discussion to the revised manuscript noting that these differences likely stem from the contrasting environmental conditions and biological communities of the two ecosystems. We also highlight that both classification approaches offer complementary insights into marine snow structure and its role in export processes.

*This is in contrast with what was observed for the Arctic system, where two successive blooms of different nature occur and are associated with different morphotypes. The first bloom was an ice edge bloom and was dominated by diatoms, while the second was ice-free and was associated with the presence of Phaeocystis, leading to agglomerated morphotypes and their slow settling compared to the first bloom (Trudnowska et al., 2021). These differences in morphotypes reflect not only contrasting environmental conditions between the Arctic and equatorial Atlantic but also distinct bloom successions and morphodynamic responses, supporting the idea that regional ecosystem characteristics shape marine snow structure and its role in export processes.*

*L434-435: In my experience, denser particles can also be more intact, larger pieces of organic detritus, or aggregated phytoplankton after a longer period of time. I think that it can't automatically be concluded that the main source of dense particles are fecal pellets.*

**Reply:** We agree that dense particles can originate from multiple sources, including intact organic detritus or aggregated phytoplankton, especially after prolonged aggregation processes. In this context, our use of the term *potential* was intended to emphasize that these dense particles are not composed solely of fecal pellets. However, we recognize that the sentence may not have conveyed this clearly. In the revised manuscript, we have rephrased this sentence to explicitly state that dense particles may include various components beyond fecal pellets, to avoid any ambiguity. The manuscript was modified as follows:

*As for dense particles, these may include fecal pellets produced by zooplankton feeding, as commonly reported in previous studies (Stemmann and Boss, 2012; Trudnowska et al., 2021), but they could also originate from other sources such as aggregated phytoplankton or phytodetritus (Alldredge and Silver, 1988; Guidi et al., 2009).*

*L482: How much of the organic matter reached those depths?*

**Reply:** We have now calculated the quantitative estimates to clarify how much organic matter reached the deep ocean. Specifically, Kiko et al. (2017) reported fluxes of approximately 2.07 mg C m$^{-2}$ d$^{-1}$ reaching depths of up to 4000 m. We have included these values in the revised manuscript to provide a clearer context for the magnitude of deep flux.

*Figure 6: Would it be possible to reflect the patterns you see in particle size in this figure?*

**Reply:** We thank Reviewer 1 for the helpful suggestion. We agree that incorporating visual cues related to particle size would improve the clarity of Figure 6. However, the figure is already quite dense, and illustrating each particle size in detail would further overcrowd it. In the revised version, we clarified in the figure's caption that the particle composition remains consistent between export and non-export events, and that the key difference lies in the concentration of particles rather than their size. While the figure already reflects these differences in concentration, we will explicitly state that both small and large particles can form dense aggregates, which are responsible for export, and that porous particles represent both the small and the large-sized particles to avoid any confusion regarding the role of particle size.

The new caption is:

*Figure 6: Illustrative example of the particle export system in the Atlantic equatorial region during export and non-export events. FP: fiber particles, PP: porous particles, DP: dense particles. Teff: Transfer efficiency, Eeff: Export efficiency, TIW: Tropical instability Wave. We do not distinguish between small and large particles, as the particle composition remains consistent across both export and non-export events and only the concentration changes.*

*Code availability*

*I think it would be helpful for the scientific community and enhance the transparency of the dataset to share the code in the supplement or on a platform such as GitHub rather than making the reader rely on direct communications with the authors.*

**Reply:** We agree with Reviewer 1; the code will be made available on GitHub to ensure accessibility and reproducibility.

---

## Author Comment (AC2)

**Marine snow surface production and bathypelagic export at the Equatorial Atlantic from an imaging float**

Joelle Habib, Lars Stemmann, Alexandre Accardo, Alberto Baudena, Franz Philip Tuchen, Peter Brandt, and Rainer Kiko

**Responses to the Reviewers' Comments**

Answers to reviewers' comments are reported point by point. The questions and comments of the reviewers are in blue, the answers in black, and the modifications that we made in the revised manuscript in red.

**Responses to the comments of the anonymous Reviewer 1**

First, we would like to warmly thank Reviewer 1 for their relevant and constructive comments, which helped to improve the manuscript.

The study provides a comprehensive data set, observing and describing two export events in the equatorial Atlantic and key factors that may drive the observed carbon flux. They present details on particle properties such as their size and density. I think that is a valuable case study of how recent developments in underwater imaging technology can be used to gain detailed insights into carbon flux mechanisms. The data analysis and method description are clear and thorough.

**Reply:** We appreciate the positive assessment of Reviewer 1.

*L44–56*: *Mineral ballasting could be mentioned as an additional factor*

**Reply:** This was added as follows in the revised manuscript.

Mineral ballasting, through the association of marine snow with dense inorganic materials such as calcium carbonate, lithogenic or biogenic silica, can also significantly enhance sinking velocities and control the carbon export efficiency (Armstrong et al., 2002; Klaas and Archer, 2002).

L315: "best transfer efficiency" – rephrase to "highest transfer efficiency" to avoid a misleading qualitative assessment

**Reply:** This was rephrased as suggested in the revised manuscript.

Section 3.4.1. I found this section difficult to read as five acronyms were introduced at once. In many points the authors make, there is a pattern by either size or by packaging. I would suggest spelling this out by describing the categories as either small or large, and either densely packed, loosely packed, or fiber. Also, I think that the word "porous" is misleading; I would suggest "dense" and "loose" or "dark" and "light", based on the image property. Example: "They shared similar temporal dynamics primarily in the surface layer: all types decreased exponentially between 0 and 150 m. While fibers and loose/light particles decreased slowly throughout the water column in the mesopelagic layers, dense/dark particles increased gradually between 400-600 m."

Reply: We thank the reviewer for this thoughtful comment. While we understand the concern about introducing multiple acronyms at once, we believe that the classification is important for

distinguishing the five morphotypes, which reflect distinct features relevant to our analysis. We recognize, however, that the section may benefit from improved clarity. Rather than replacing the acronyms entirely, we prefer to retain them but will ensure that each is introduced with accompanying descriptive terms (e.g., small/large, dense/loose, or fiber-like) to improve readability.

Regarding the use of the term "porous," we acknowledge that interpretation can vary. However, we use this term to reflect specific textural characteristics observed in the images, and we are cautious about replacing it with terms like "loose", which may carry a different meaning. We remain open to refining the terminology further if needed during the revision process.

The k-means clustering applied to the PCA coordinates helped us to distinguish between five marine snow morphotypes, illustrated in Figure 5. Type 1 consisted of large, compact objects with an Equivalent Spherical Diameter (ESD)>0.8mm, referred to as Big Dense Particles (BDP). Type 2 comprised elongated, thread-like objects termed Fiber particles (FP), and type 3 consisted of large bright, and porous objects referred to as Big Porous particles (BPP). Type 4 was mainly formed of dense, small, and circular objects: Small Dense particles (SDP), and type 5 consisted of small, bright, and porous objects: Small Porous particles (SPP). These five morphotypes were then used to characterize the distribution and composition of marine snow. It should be noted that the terms "porous" and "dense" refer to brightness, with "porous" indicating greater light transmission while "dense" denotes lower light transmission.

The morphotypes identified here are different than the clusters identified by Trudnowska et al (which are cited here). Discussing why different categories were identified in the context of this ecosystem would add value to the paper.

Reply: We agree that addressing the differences between our morphotype categories and the clusters identified by Trudnowska et al. (2021) would strengthen the manuscript. Trudnowska et al. categorized marine snow into five broad morphotypes (dark, elongated, flake, fluffy, and agglomerated), optimized for the Arctic environment, particularly to capture the high morphological variability across bloom phases and changing phytoplankton communities in marginal ice zones. Their emphasis was on temporal structural changes concerning Arctic bloom dynamics. In contrast, our study, conducted in the equatorial Atlantic, defines five morphotypes that we interpret as reflecting morphodynamic features specific to this region's ecological and physical context. In addition, they have been using the previous version of the camera (UVP5), whose image properties are slightly different, making direct comparisons of images difficult, while qualitative comparisons between the two studies are possible. We have now added a brief discussion to the revised manuscript, noting that these differences likely stem from the contrasting environmental conditions and biological communities of the two ecosystems. We also emphasize that both classification approaches provide complementary insights into the structure of marine snow and its role in export processes.

This is in contrast with what was observed for the Arctic system, where two successive blooms of different nature occur and are associated with different morphotypes. The first bloom was an ice-edge bloom and was dominated by diatoms, while the second was ice-free and was associated with the presence of Phaeocystis, leading to agglomerated morphotypes and their slow settling compared to the first bloom (Trudnowska et al., 2021). These differences in morphotypes reflect not only contrasting environmental conditions between the Arctic and equatorial Atlantic but also distinct bloom successions and morphodynamic responses, supporting the idea that regional ecosystem characteristics shape marine snow structure and its role in export processes.

L434-435: In my experience, denser particles can also be more intact, larger pieces of organic detritus, or aggregated phytoplankton after a longer period of time. I think that it can't automatically be concluded that the main source of dense particles are fecal pellets.

**Reply:** We agree that dense particles can originate from multiple sources, including intact organic detritus or aggregated phytoplankton, especially after prolonged aggregation processes. In this context, our use of the term *potential* was intended to emphasize that these dense particles are not composed solely of fecal pellets. However, we recognize that the sentence may not have conveyed this clearly. In the revised manuscript, we have rephrased this sentence to explicitly state that dense particles may include various components beyond fecal pellets, to avoid any ambiguity. The manuscript was modified as follows:

As for dense particles, these may include fecal pellets produced by zooplankton feeding, as commonly reported in previous studies (Stemmann and Boss, 2012; Trudnowska et al., 2021), but they could also originate from other sources such as aggregated phytoplankton or phytodetritus (Alldredge and Silver, 1988; Guidi et al., 2009).

**L482: How much of the organic matter reached those depths?**

**Reply:** We have now calculated the quantitative estimates to clarify how much organic matter reached the deep ocean. Specifically, Kiko et al. (2017) reported fluxes of approximately 2.07 mg C m-2 d-1 reaching depths of up to 4000 m. We have included these values in the revised manuscript to provide a clearer context for the magnitude of deep flux.

**Figure 6: Would it be possible to reflect the patterns you see in particle size in this figure?**

**Reply:** We thank Reviewer 1 for the helpful suggestion. We agree that incorporating visual cues related to particle size would improve the clarity of Figure 6. However, the figure is already quite dense, and illustrating each particle size in detail would further overcrowd it. In the revised version, we clarified in the figure's caption that the particle composition remains consistent between export and non-export events, and that the key difference lies in the concentration of particles rather than their size. While the figure already reflects these differences in concentration, we will explicitly state that both small and large particles can form dense aggregates, which are responsible for export, and that porous particles represent both the small and the large-sized particles to avoid any confusion regarding the role of particle size.

The new caption is as follows:

Figure 6: Illustrative example of the particle export system in the Atlantic equatorial region during export and non-export events. FP: fiber particles, PP: porous particles, DP: dense particles. Teff: Transfer efficiency, Eeff: Export efficiency, TIW: Tropical instability Wave. We do not distinguish between small and large particles, as the particle composition remains consistent across both export and non-export events and only the concentration changes.

**Code availability**

I think it would be helpful for the scientific community and enhance the transparency of the dataset to share the code in the supplement or on a platform such as GitHub rather than making the reader rely on direct communications with the authors.

**Reply:** We agree with Reviewer 1; the code will be made available on GitHub to ensure accessibility and reproducibility.

**Responses to the comments of the anonymous Reviewer 2**

First, we would like to warmly thank Reviewer 2 for their relevant and constructive comments, which helped to improve the manuscript.

The authors investigate two bloom events and associated particle/carbon export dynamics in the equatorial Atlantic using BGC Argo data and remote sensing products. Identification of bloom characteristics, potential drivers, and mechanisms determining export of biogenic particles during and outside these two events are presented in a largely convincing manner. The manuscript presents an interesting case study that nicely highlights the value of UVP data for more in-depth assessments of the composition and fate of organic matter associated with phytoplankton blooms. The manuscript is largely well written but may benefit from minor language editing. I have a few general and minor comments. Once these have been addressed, I would recommend the manuscript for publication.

**Reply:** We appreciate the positive assessment of Reviewer 2.

The criterion used to constrain the particle plumes, i.e. sloped lines corresponding to a particle sinking speed of 30 m day-1, does not seem to adequately capture the second plume (Fig. 4d). However, the actual sinking speed of particles in this plume seem to be higher than the literature value of 30 m day-1 (which could be explained by the higher concentration of large particles). Wouldn't it make sense to adjust the criterion to get better-fitting constraints for the plume and subsequent analysis?

**Reply:** We thank the reviewer for this valuable observation. The criterion of 30 m day-1 was chosen based on commonly reported literature values for particle sinking speeds. In fact, the plumes contain different types of sinking particles with different sinking speeds, so the signature of the plume changes according to the types of particles, and one may consider that we have as many plumes as we have particle types. We agree that the second plume in Fig. 4d appears to exhibit faster sinking velocities, likely due to a higher abundance of large, rapidly sinking particles. However, because we show that particle composition does not change between the plumes and to maintain consistency across all analyzed events, we kept the same 30 m day-1 criterion.

We have clarified this reasoning in the revised manuscript and noted the possibility that the actual sinking speed in this specific plume may be higher.

Line 288-297

To investigate two settling plumes depicted in Figure 3d, four spaced lines were drawn on the MaP abundance with a slope of 30 m day-1, as suggested by Stemmann, Jackson, & Gorsky (2004) using a model for particle size distribution. In reality, the plumes contain different types of sinking particles with varying velocities. Although the second plume appears to exhibit a higher sinking speed, particle

composition does not change between the plumes. Therefore, to maintain consistency across all events, the same 30 m day-1 criterion was applied throughout. The periods of surface production and export were determined using the Rodionov algorithm (Fig. S4). The first export event, "Event 1", started at the surface on the 8th of August and lasted until the 8th of September 2021, while the second event, "Event 2", occurred from the 13th of December 2021 to the 26th of January 2022. Both events lasted one month (Fig. 3f). These events are easily discernible on the MaP abundance and carbon flux plots as two plumes that reach 2000 m depth (Fig. 3c-f) while they are less visible in the MiP pattern.

The carbon flux units mentioned throughout the manuscript and figures need to be revised. Units are given in mgC m-3 d-1 in the text (e.g., lines 279, 281) and in mgC m-3 in the figure labels (Figs. 3 and 4). However, fluxes should be given in mass per area per time, i.e., mgC m-2 d-1.

**Reply:** We apologize for the oversight. The carbon fluxes are indeed in mgC m-2 d-1. We have corrected the units throughout the text and figures to reflect mass per area per time consistently.

In the discussion on diel vertical migration as an explanation for the increase in small dense particles at depth, I suggest mentioning the local time of day at which the float profiled (which would also be helpful information to include in the methods). According to the Argo fleet monitoring website, profiles were all taken near local midnight, which would support the hypothesis that the increased abundance in particles at depth are a diel migratory signal.

**Reply:** We thank the reviewer for this helpful suggestion. Indeed, 50% of the profiles were conducted during the night. We have added this information to the Methods section and now mention it in the Discussion to support the interpretation that the observed increase in small dense particles at depth may be linked to diel vertical migration.

Line 143-150

An Underwater Vision Profiler 6 (UVP6) was mounted on the BGC Argo float. This camera-based particle counter sizes and counts marine particles (Kiko et al., 2022) covering a size range from 0.102 mm to 16.4 mm. The UVP contributes to understanding sinking organic particles and carbon sequestration at global (Guidi et al., 2015) and regional scales (Ramondenc et al., 2016). More information about calibration and data processing can be found in Picheral et al. (2021). In total, our data set includes 86 profiles reaching at least 1000 m. Every 3 days, the BGC Argo float reached 2000 m. According to the recorded profiling times, at least 50% of the profiles were conducted during nighttime, with many occurring close to local midnight. For all parameters, we interpolated the data set with a vertical resolution of 10 m and a temporal resolution of a day.

Line 100: Why were different spatial resolutions used for the two satellite products? VIIRS should be available in 4km resolution as well, right?

**Reply:** We appreciate the reviewer's comment. Several (partly overlapping) satellite missions have measured ocean color data since 1997 and are used to infer surface Chl-a concentration. Because cloud coverage prevents visible-spectrum satellite measurements, data gaps are common in the tropical rain belt. To obtain continuous, gap-free time series, multi-sensor products have been developed that merge data from several missions and apply data-interpolating empirical orthogonal functions. In this study, we used two such products: the **Copernicus GlobColour** daily composite of

surface Chl-a concentration, available since 1997 at **4 km** horizontal resolution (Copernicus, 2023b), and the **NOAA Multi-Sensor** (**MSL12**) CoastWatch product, available since 2018 at **9 km** resolution (NOAA CoastWatch). Both datasets provide daily, gap-free fields but differ in their gap-filling methodologies, input satellite missions, and record lengths. These methodological differences lead to distinct effective spatial resolutions in the final products. In general, the finer (4 km) resolution product likely carries higher uncertainty, as it requires more extensive interpolation in regions with frequent cloud cover.

**Line 137: What wavelength was BBP measured at? 470 or 700 nm?**

**Reply:** BBP was measured at 700 nm. We have added this information to the Methods section.

Line 136-137:

This float was equipped with several physical and biogeochemical sensors to measure the pressure, temperature, salinity, chlorophyll, oxygen, and particle backscattering coefficient (BBP, measured at 700 nm) with a vertical resolution of 5 m.

**Line 152: Is 60m the maximum mixed layer depth that was reached?**

**Reply:** During that period, and based on the mixed layer depth defined as a decrease of 0.2 °C relative to the temperature at 10 m depth, the mixed layer reached a maximum depth of 60 m in this region.

Line 252-254: Another factor that would contribute to the difference in variability: Float and satellite chl are derived in fundamentally different ways. Float chl is derived from fluorescence, which is affected by physiology (primarily by light and nutrient availability). Satellite chl is derived from remote sensing reflectance, which is largely unaffected by physiology. Float-based chl estimates are a lot more variable for this reason. See Long et al. (2024), https://www.nature.com/articles/s43247-024-01762-4

**Reply:** We thank the reviewer for this insightful comment. We agree that differences in the derivation of chl-a between float and satellite data contribute to the observed variability. Float chl-a is derived from fluorescence and is influenced by physiological factors such as light and nutrient availability, whereas satellite chl-a is derived from remote sensing reflectance and is largely unaffected by physiology (Long et al., 2024). We have updated the manuscript to include this explanation, in addition to the previously mentioned factors of lower satellite resolution and interpolation methods.

**Lines 252-254**

However, float data presented a more variable chl-a concentration compared to satellite data. This can be attributed not only to the lower spatial resolution of satellite images and the interpolation methods used to ensure a gap-free time series, but also to fundamental differences in how chl-a is measured. Float-based chl-a is derived from fluorescence and is sensitive to phytoplankton physiology, community composition, nutrient availability, and environmental conditions, whereas satellite measurements rely on optical reflectance and are largely insensitive to these factors (Long et al., 2024).

**Fig. 3: Are the data shown here log10 or natural log transformed?**

**Reply:** The data shown in Fig. 3 are neither log10 nor natural log transformed. The values represent the abundances of small and large particles, and the carbon flux was calculated using the equation from Kriest et al. (2002). The anomalies shown correspond to deviations from the mean profiles.

*Line 340: I assume this is a typo: "SDP were most abundant ..." (instead of SPP).*

Reply: We apologize for the typo and have corrected "SDP" to "SPP" in the revised manuscript.

Line 417: Correct me if I'm wrong, but this seems to contradict what's stated in the results. Here: "Both MiP and MaP in the top 100 n are correlated to in situ chl-a biomass". In line 269: "No significant correlation was found between surface chl-a and MaP abundance."

**Reply:** Thank you. This was a mistake on our part. To clarify, MiP and MaP integrated over the upper 100 m are significantly correlated with in-situ chl-a biomass (Fig. S3). By contrast, no significant correlation was found between MaP abundance measured at the near-surface sampling depth and the near-surface chl-a. We have corrected the text to remove the inconsistency.

Line 275-278

The increase in surface chl-a in the first 100 m was linked to an increase in surface carbon flux, MaP and MiP abundance (particles between 0.1-0.5 mm) (Fig. S3). All were significantly correlated with in situ chl-a ( $r^2$ =0.4, 0.6, and 0.3, respectively, p-value<0.01). No significant correlation was found between near-surface chl-a and MaP abundance (particles>0.5 mm).

**Fig. S2: Consider adding SD shades behind the climatology.**

**Reply:** We agree with this suggestion. Standard deviation shading has been added behind the climatology curves for panels A and B in Fig. S2.

Fig. S7: Colours for FD and BPP are not colourblind-friendly.

**Reply:** We agree that the previous color scheme was not optimal for colorblind accessibility. We have updated the colors for FD and BPP in Fig. S7 using a colorblind-friendly palette to improve readability and accessibility.

Fig. S9: The labels are somewhat cryptic without an explanation of what the variables mean.

**Reply:** We agree that additional clarification is needed. We have revised the figure caption to include definitions of all variables and abbreviations used in Fig. S9. In addition, we have updated the corresponding table labels. The figure was modified as follows:

Figure S9. Correlations of the carbon flux, MiP, MaP, and cluster abundance in the first 100 m with the Lagrangian diagnostics and in situ data. Cluster 1: BDP; Cluster 2: FP; Cluster 3: BPP; Cluster 4: SDP; and Cluster 5: SPP. Ftle: Finite-Time Lyapunov Exponents; Betw: Betweenness; Lag: Lagrangian; Eul: Eulerian; Div: divergence; Vort: vorticity; Chl: chlorophyll; Prec: precipitation.